# Metapopulation dynamics of SARS-CoV-2 transmission in a small-scale Amazonian society

Thomas S. Kraft [1,2,3] *, Edmond Seabright[4,5], Sarah Alami[2,4], Samuel M. Jenness[6], Paul Hooper[7], Bret Beheim[3], Helen Davis[8], Daniel K. Cummings[7], Daniel Eid Rodriguez[9], Maguin Gutierrez Cayuba[10], Emily Miner[2], Xavier de Lamballerie[11], Lucia Inchauste[11], Stéphane Priet[11], Benjamin C. Trumble[12,13], Jonathan Stieglitz[14], Hillard Kaplan[7‡], Michael D. Gurven[2‡]

1 Department of Anthropology, University of Utah, Salt Lake City, Utah, United States of America,
2 Department of Anthropology, University of California Santa Barbara, Santa Barbara, California, United States of America, 3 Department of Human Behavior, Ecology, and Culture, Max Planck Institute for Evolutionary Anthropology, Leipzig, Germany, 4 School of Collective Intelligence, Mohammed VI Polytechnic University, Rabat, Morocco, 5 University of New Mexico, Department of Anthropology, Albuquerque, New Mexico, United States of America, 6 Department of Epidemiology, Emory University, Atlanta, Georgia, United States of America, 7 Department of Health Economics and Anthropology, Economic Science Institute, Argyros School of Business and Economics, Chapman University, Orange, California, United States of America, 8 Department of Human Evolutionary Biology, Harvard University, Cambridge, Massachusetts, United States of America, 9 Universidad Mayor de San Simon, Cochabamba, Bolivia, 10 Tsimane Gran Consejo, San Borja, Bolivia, 11 Unité des Virus Émergents (UVE: Aix-Marseille Univ–IRD 190 –Inserm 1207 –IHU Méditerranée Infection), Marseille, France, 12 School of Human Evolution and Social Change, Arizona State University, Tempe, Arizona, United States of America, 13 Center for Evolution and Medicine, Arizona State University, Tempe, Arizona, United States of America, 14 Institute for Advanced Study in Toulouse, Toulouse, France

‡ These authors are joint senior authors on this work.
* thomas.kraft@utah.edu

**Data Availability Statement:** Code and data underlying the figures and analysis associated with this paper are available at http://doi.org/10.17605/OSF.IO/7YB2M. Empirical data on the Tsimane

## Abstract

The severity of infectious disease outbreaks is governed by patterns of human contact, which vary by geography, social organization, mobility, access to technology and health-care, economic development, and culture. Whereas globalized societies and urban centers exhibit characteristics that can heighten vulnerability to pandemics, small-scale subsistence societies occupying remote, rural areas may be buffered. Accordingly, voluntary collective isolation has been proposed as one strategy to mitigate the impacts of COVID-19 and other pandemics on small-scale Indigenous populations with minimal access to healthcare infra-structure. To assess the vulnerability of such populations and the viability of interventions such as voluntary collective isolation, we simulate and analyze the dynamics of SARS-CoV-2 infection among Amazonian forager-horticulturalists in Bolivia using a stochastic network metapopulation model parameterized with high-resolution empirical data on population structure, mobility, and contact networks. Our model suggests that relative isolation offers little protection at the population level (expected approximately 80% cumulative incidence), and more remote communities are not conferred protection via greater distance from out-side sources of infection, due to common features of small-scale societies that promote rapid disease transmission such as high rates of travel and dense social networks. Neigh-borhood density, central household location in villages, and household size greatly increase

population used as input for the model are highly identifiable (GPS locations of households, individual household sizes, ages, and sexes, etc.) and thus are not publicly available. Individual-level data are stored in the Tsimane Health and Life History Project (THLHP) Data Repository, and are available through restricted access for ethical reasons. THLHP's highest priority is the safeguarding of human subjects and minimization of risk to study participants. The THLHP adheres to the "CARE Principles for Indigenous Data Governance" (Collective Benefit, Authority to Control, Responsibility, and Ethics), which assure that the Tsimane 1) have sovereignty over how data are shared, 2) are the primary gatekeepers determining ethical use, 3) are actively engaged in the data generation, and 4) derive benefit from data generated and shared for use whenever possible. The THLHP is also committed to the "FAIR Guiding Principles for scientific data management and stewardship" (Findable, Accessible, Interoperable, Reusable). Requests for individual-level data should take the form of an application that details the exact uses of the data and the research questions to be addressed, procedures that will be employed for data security and individual privacy, potential benefits to the study communities, and procedures for assessing and minimizing stigmatizing interpretations of the research results (see the following webpage for links to the data sharing policy and data request forms: https://tsimane.anth.ucsb.edu/data.html). Requests for individual-level data will require institutional IRB approval (even if exempt) and will be reviewed by an Advisory Council composed of Tsimane community leaders, community members, Bolivian scientists, and the THLHP leadership. The study authors and the THLHP leadership are committed to open science and are available to assist interested investigators in preparing data access requests.

**Funding:** Funding is from National Institute of Health (NIH)/National Institute on Aging (NIA) grant RF1AG054442 to MG, HK, JS, and BT. JS acknowledges funding from the French National Research Agency under the Investments for the Future (Investissements d'Avenir) programme (ANR-17-EURE-0010). The funders had no role in study design, data collection and analysis, decision to publish, or preparation of the manuscript.

**Competing interests:** The authors have declared that no competing interests exist.

**Abbreviations:** COVID-19, Coronavirus Disease 2019; GLMM, generalized linear mixed model; IRB, institutional review board; MCMC-MLE, Markov Chain Monte Carlo maximum likelihood estimation;

the individual risk of infection. Simulated interventions further demonstrate that without implausibly high levels of centralized control, collective isolation is unlikely to be effective, especially if it is difficult to restrict visitation between communities as well as travel to outside areas. Finally, comparison of model results to empirical COVID-19 outcomes measured via seroassay suggest that our theoretical model is successful at predicting outbreak severity at both the population and community levels. Taken together, these findings suggest that the social organization and relative isolation from urban centers of many rural Indigenous communities offer little protection from pandemics and that standard control measures, including vaccination, are required to counteract effects of tight-knit social structures characteristic of small-scale populations.

## Introduction

Indigenous populations worldwide share certain characteristics that elevate vulnerability to infectious disease outbreaks [1]. This vulnerability has historically manifested as greater relative mortality rates compared to non-Indigenous populations during epidemics, including measles, influenza, and malaria in Amazonia after contact with Europeans [2], the 1918 influenza pandemic among Maori, Arctic, and Pacific peoples [3,4], and the 2009 H1N1 influenza pandemic among Aboriginal Australians, Pacific Islanders, Maori, First Nations peoples, and Alaska Natives [5]. Factors increasing vulnerability of Indigenous populations include complications from previous exposures to respiratory diseases, comorbidities, adverse socioeconomic conditions, minimal access to health and sanitation infrastructure, and discrimination in local healthcare systems [6–10]. Due to these and other factors, Indigenous communities worldwide have also suffered disproportionately during the ongoing Coronavirus Disease 2019 (COVID-19) pandemic from especially high morbidity and mortality [11–16].

Data-driven research is needed to guide effective interventions and public health strategies in Indigenous communities during pandemics. Ideal strategies would account for the particular features of social structure, geographical distribution, and contact that characterize such populations. For example, during the global spread of Severe Acute Respiratory Syndrome Coronavirus 2 (SARS-CoV-2), specific efforts were made to mitigate viral transmission and impact in Indigenous Tsimane communities of the Bolivian Amazon. This effort raised key questions regarding how the disease might spread in remote, small-scale populations and whether a multistage plan emphasizing a prevention strategy of voluntary collective isolation (self-isolation at the group level promoting limited interaction with outsiders) followed by contact tracing and a targeted distribution of available medical resources could be effective [6]. For example, which features of Indigenous communities (e.g., size, density, location) render them most vulnerable to COVID-19? How much safer are remote communities than communities located near market towns? How is COVID-19 likely to spread once it has reached rural communities? Are certain subgroups (e.g., by age or sex) more likely to be exposed and transmit disease? How strict must voluntary collective isolation be in order to succeed? Should travel restrictions extend to within the Indigenous territory, and to the most remote regions? Controlling disease outbreaks among Indigenous communities is complicated by a paucity of detailed information on social organization and mobility, and socioecological features of these populations that poorly fit the assumptions of standard epidemiological models commonly used in urban, industrialized contexts [17,18].

Contact networks and thus disease transmission dynamics are directly influenced by certain demographic, organizational, and political features shared by many small-scale societies, such

SARS-CoV-2, Severe Acute Respiratory Syndrome Coronavirus 2; SEIRD, susceptible-exposed-infectious-recovered-death; TERGM, temporal exponential-family random graph model; THLHP, Tsimane Health and Life History Project.

as "bottom-heavy" age pyramids, close residential proximity of intergenerational households coupled with communal living that facilitates transfers of food and other resources, and relatively egalitarian decision-making at the group level (e.g., [19]). The metapopulation structure of some groups—consisting of separate villages connected via kinship, visitation, and trade, and with variable contact patterns with "outsiders" (non-Indigenous individuals living outside Indigenous territories)—differs markedly from that of large-scale urban contexts. Standard epidemiological models, particularly deterministic compartmental models that assume sufficiently large, well-mixed populations [20], are therefore unlikely to be useful for guiding public health decisions in many small-scale Indigenous communities that are relatively isolated from major urban centers. Fortunately, there is now widespread recognition that the structure of contact networks affects epidemiological outcomes [20] and new mathematical models have been developed that are capable of representing underlying network structures (e.g., [21–23]) and spatial organization [24]. These individual-based models integrate biological and social phenomena to investigate the underlying mechanisms driving infectious disease transmission dynamics [25] across diverse social and environmental contexts [26].

To explore the dynamics of infectious disease transmission and potential intervention strategies in rural, small-scale, Indigenous societies, we developed an individual-based stochastic network model that incorporates realistic features derived from long-term, longitudinal empirical data from one Indigenous population, the Tsimane forager-horticulturalists of lowland Amazonian Bolivia (Fig 1) [27]. The Tsimane are a largely autonomous subsistence population inhabiting a territory outside of urban centers; their social organization can be

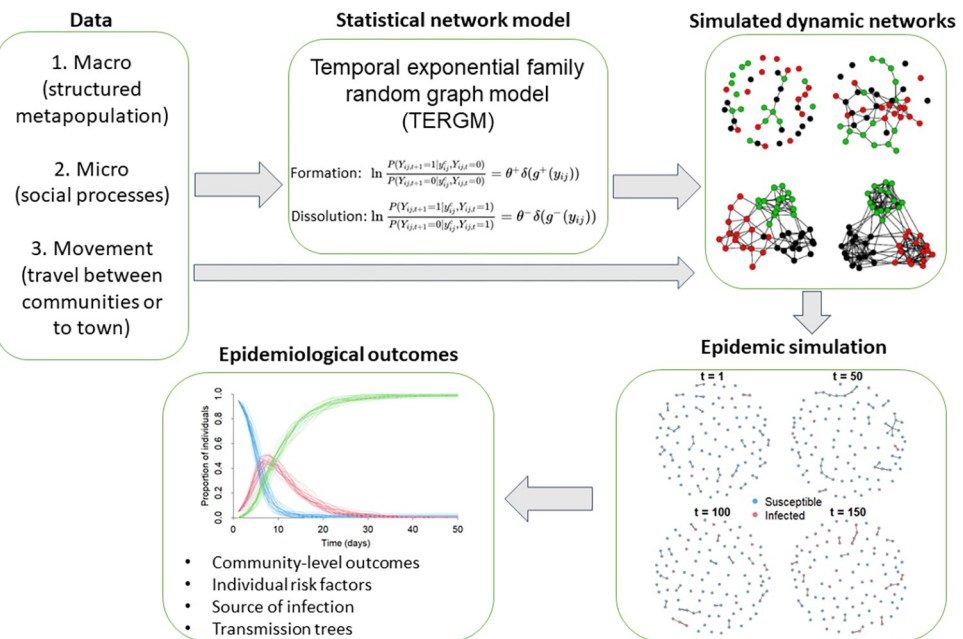

**Fig 1. Diagram of the modeling procedure.** Empirical data on macro- and microlevel social processes describing the study population are used to fit a generative TERG model. When dynamic networks are simulated from non-degenerate TERGMs, resulting network properties stochastically reproduce target statistics (age/sex homophily, interactions with genetic kin, etc.) in expectation. Custom modules are applied in conjunction with network simulation to implement SEIRD transitions and move individuals between villages and to town. Zero cases are seeded in the population at the beginning of each simulation, with initial infectious disease seeding occurring during travel to town. Infections are transmitted probabilistically based on contacts in resulting networks and input parameters. Individual outcomes are tracked in a transmission matrix allowing for postsimulation analysis.

characterized as a dispersed metapopulation of tightly knit, kin-based communities in a rural region with limited access to modern medical resources. This suite of characteristics is common to a broad range of Indigenous societies globally, making the well-described Tsimane case study a useful reference for understanding infectious disease dynamics and effective intervention strategies in other Indigenous populations.

We begin by drawing on extensive data collected over the past 2 decades to characterize Tsimane social networks, mobility, spatial structure, and demography. The resulting high-resolution description of this system is then used to parameterize a dynamic metapopulation network model representing the complete adult population of Tsimane living in 65 villages ($n$ = 7,269 individuals). Networks evolve dynamically over the course of simulation, with mobility parameters governing visitation to market towns and travel between Tsimane communities. We use parameters that reflect the characteristics of SARS-CoV-2 to simulate the introduction (from contact with urban Bolivians) and spread of disease among Tsimane. We evaluated (1) how socioecological features of an Indigenous small-scale society influence the extent and trajectory of infectious disease (COVID-19) spread at the population-level; (2) community- and individual-level risk factors for susceptibility to infection; and (3) the effect of potential interventions (i.e., travel restrictions either to town or between villages, altering disease transmissibility in towns or within villages [e.g., via facial coverings] and restricting within-village gatherings) on the final outbreak size and trajectories of epidemics. Finally, we compare our model results to observed outcomes based on seroassays from 612 Tsimane individuals measured after a first wave of COVID-19 infection in this population, assessing outcomes at the population level, by sex, and by community.

## Results

We first present results on total outbreak size by describing cumulative incidence of infection at the population, community, and individual levels from our baseline model, which adopts standard parameters associated with SARS-CoV-2. We then examine disease trajectories by examining the timing of infections, with special attention to explaining variability among communities. Finally, we examine the efficacy of several potential intervention strategies by examining how epidemiological outcomes respond to changes in model parameters.

### Total outbreak size

**Population level.** Our baseline model predicts extremely high cumulative incidence (mean [95% percentile interval] = 80.9 [79.2, 82.3]) and relatively few deaths (mean [95% CI] = 31.4 [22.0, 38.5]) at the population level after the epidemic runs its course over 150 days and implementing a relatively modest transmissibility parameter (Fig 2A and Table 1). Epidemiological trajectories at the population level varied little across model runs despite a heterogeneous metapopulation and stochasticity in both network formation and movement patterns. Although the trajectory of disease spread appears to track the artificially imposed profile of infection probability at the source (town), the observed exponential increase in infections at early time points is driven almost entirely by within-village transmission, seeded through travel to town or other villages where the epidemic was already introduced (Fig 3). A visual example of this process in a single community is available as a supplementary animation (S2 Fig) in which a single individual is exposed at $t$ = 17–18 while traveling to town, subsequently sparking a local epidemic.

**Community level.** Median outbreak size at the community level (proportion infected after 150 simulation days) was 0.83 (SD = 0.08, range = 0.42–1.0) across all model simulations (S1 Fig). Under baseline conditions, no communities consistently avoided introduction of the

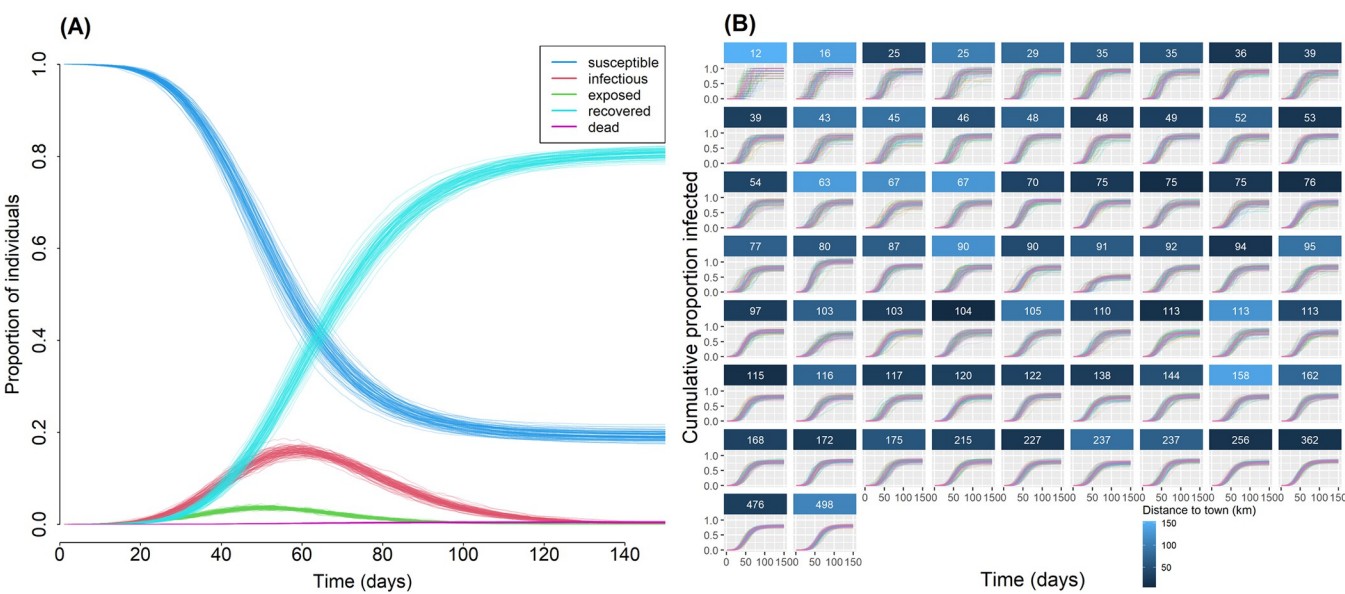

**Fig 2.** Baseline scenario model results for (**A**) the proportion of individuals in SEIRD compartments over time, and (**B**) cumulative proportion of infected individuals by community. Communities in (**B**) are ordered according to increasing community size (labeled by panel) and are colored according to distance of community from town. The data underlying this Figure can be found in http://doi.org/10.17605/OSF.IO/7YB2M (files: https://osf.io/jb5x7, https://osf.io/yar47, https://osf.io/p2shn).

disease across simulations (Fig 2B). This suggests that even the smallest, most remote communities are at significant risk. Although smaller communities had more variable disease trajectories than larger communities (Figs 2B and S1), smaller communities consistently experienced more severe outbreaks compared to larger communities controlling for density, community centrality, and distance to town (β [95% CI] = −0.270 [−0.39, −0.16]; Table 2: column 1 and Fig 4B). Distance to the nearest market town, average community density, and community betweenness centrality (based on the travel network), in contrast, had no discernable effect on community outbreak size (Fig 4A–4D and Table 2).

**Individual level.** Models of individual infection probability largely recapitulate community-level results but offer the opportunity to examine individual attributes (e.g., age, sex, household size, local neighborhood density). Adjusting for community properties, including community size, distance to town, and average community density, probability of individual infection by the end of simulation was associated with all individual predictors tested: Individuals who were older, female, lived in a more densely populated neighborhoods closer to the center of the community, and in bigger households had a higher probability of infection by the end of the simulation (Table 3: Model 1) and experienced earlier infection times (Fig 5 and Table 3: Model 2). Of these factors, neighborhood density and household size had by far the largest standardized effect sizes (0.39 and 0.73, respectively), whereas age and sex effects were relatively weak (Fig 5A and 5B versus Fig 5C–5F).

## Disease trajectories: Timing of onset and maximal spread

**Population level.** Trajectories of exposure and infection varied little across model simulations despite large variation in where local outbreaks occurred first (Fig 2). The number of infected individuals in the population peaked consistently around days 50 to 60, which is near the point at which infections due to town visitation were diminishing (S4 Fig).

**Table 1. Input parameters for the baseline model.**

| Parameter | Description | Value | Source/rationale |
|---|---|---|---|
| Transmissibility | Probability of infection given one contact between susceptible and infectious individual. | 0.05 | This parameter poorly known given that it varies based on local environment (outdoor vs. indoor, humidity, etc.), the nature of spread (aerosol, fomite, direct contact), and tremendous variability in the nature of interpersonal interaction (e.g., repeated contact within a household vs. transient interaction). The value specified corresponds with the mid-lower end of the range of the pooled estimate reported in the meta-analysis of SARS-CoV-2 secondary attack rate [60] and is similar to the secondary attack rate reported for some studies in relatively similar environments [61]. See S1 Text section "Transmissibility parameter" for more details. |
| Contact rate | Number of repeated contacts per day | 1 | Arbitrarily set such that transmissibility equals effective contact rate. |
| Exposed (noninfectious) duration | Number of expected days between exposed and infectious states. | 3 | Meta-analysis of studies in China [62], showing a median incubation period of 5 days and assuming that individuals are infectious a few days prior to showing symptoms. |
| Infection duration | Number of expected days between entering infectious state and recovery or death. | Mean = 10 days | [63] CDC guidance suggesting individuals remain infectious up to 10 days: (https://www.cdc.gov/coronavirus/2019-ncov/hcp/duration-isolation.html) |
| Age-specific case fatality rate | Probability of death instead of recovery at end of infectious period, by age. | See source | IFR values from Table 1 in [59] |
| Town infection risk | Probability of exposure during a single time step (1 day) in town | Truncated normal distribution (max = 0.05, min = 0, start = day 0, end = day 60); see S4 Fig | Based on single-wave infection dynamics, with maximum corresponding to effective contact rate within-community. |
| Town visit duration | Length of time individuals stay in town when traveling. | 1 | Based on observation that many trips occur for brief economic reasons. |
| Intercommunity visit duration | Length of time individuals stay in nonresident communities when traveling. | 1 | Simplifying assumption, such that the overall probability of intercommunity travel is matched on average by sampling travel probability each day. |

**Community level.** In contrast to total outbreak size, the shapes of disease trajectories were influenced by several community-level variables. The time of first infection, which describes how early the disease arrived in a particular community, was later for communities that were smaller, farther from town, and less densely populated (Fig 4E–4H and Table 2: column 2). This result is consistent with the observation that people from communities near towns travel to those towns at greater frequency and also reflects the greater probability of infection arriving earlier in communities with more individuals.

Likewise, the time at which the proportion of infectious individuals in a village reached maximum was earlier in communities that are closer to town and smaller in size but was unrelated to community density or centrality (Fig 4I–4L and Table 2: column 3). Holding other covariates at their means, a community in the 10th percentile for community size and distance from town would reach peak proportion infectious 11 days earlier (54 versus 65 days) than a community in the 90th percentiles for those same variables.

Finally, the maximum proportion infectious at any given time is a useful measure of the rate and intensity of disease spread, as it indicates the per-capita number of active infections that require medical treatment or are liable to continue spreading the disease. In contrast to the time of first infection or the time of maximum proportion infectious, this outcome was strongly influenced by only community size, translating to an 11% higher peak proportion of infectious adults for a community in the 10th percentile for size relative to the 90th percentile (0.294 versus 0.187) (Fig 4M–4P and Table 2: column 4).

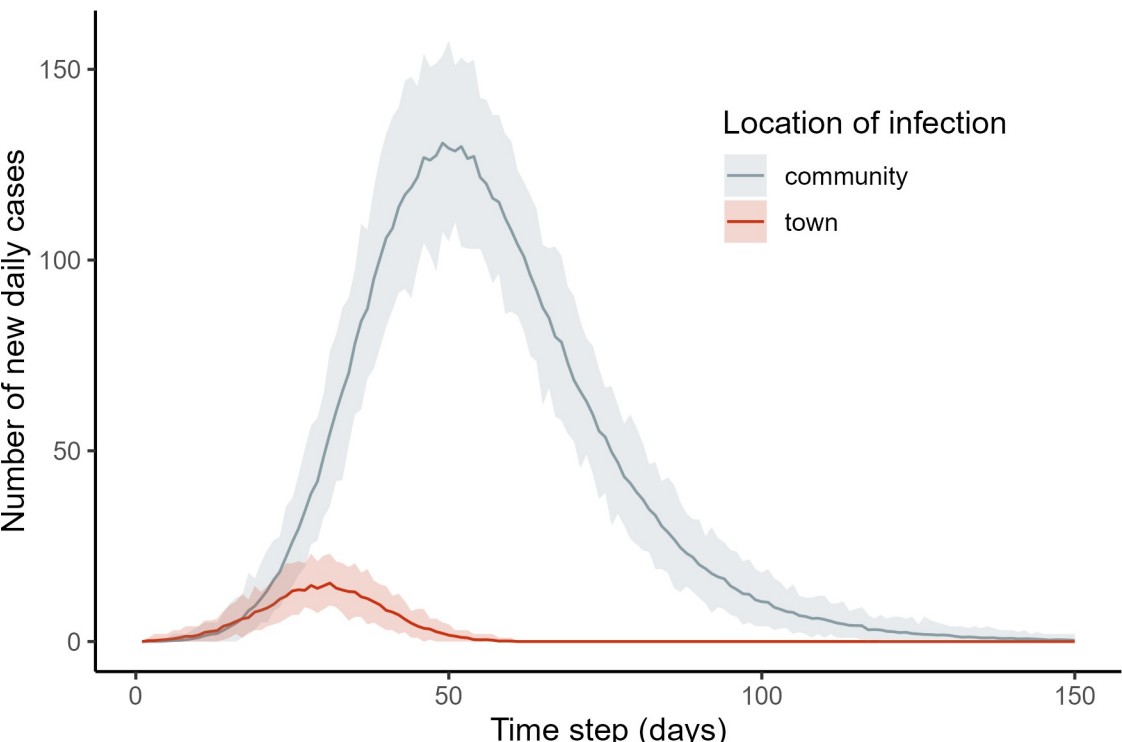

**Fig 3. Incidence of new cases contracted in town versus within-communities in the baseline model.** Lines and shaded regions represent mean and 95% percentile intervals across model runs. The data underlying this Figure can be found in http://doi.org/10.17605/OSF.IO/7YB2M (files: https://osf.io/jb5x7).

**Table 2. Regression model results for community-level predictors of total outbreak size (proportion infected), time of first infection, time of maximum proportion infectious, and maximum proportion infectious[1].** In addition to the fixed effects shown (all z-scored), the models include a random intercept effect for community across simulations. Effect sizes represent mean [95% bootstrapped CI]. Effects whose 95% CIs do not overlap with zero are bolded.

| | Dependent variable: | | | |
|---|---|---|---|---|
| | **Proportion infected** | **Time of first infection** | **Time of max proportion infectious (days)** | **Max proportion infectious at any given time** |
| | **(1)** | **(2)** | **(3)** | **(4)** |
| ln Community size (# individuals) | **−0.270** [−0.386, −.161] | **−0.150** [−0.181, −0.116] | **0.033** [0.013, 0.052] | **−0.042** [−0.049, −0.034] |
| ln Community density (mean individuals within 1 km radius) | 0.055 [−0.066, 0.170] | **−0.059** [−0.093, −0.027] | −0.019 [−0.040, 0.001] | 0.003 [−0.006, 0.010] |
| ln Community centrality (betweenness + 1) | −0.047 [−0.108, 0.015] | −0.005 [−0.022, 0.012] | 0.003 [−0.009, 0.013] | −0.003 [−0.008, 0.001] |
| Distance to town (km) | −0.005 [−0.077, 0.064] | **0.063** [0.043, 0.085] | **0.038** [0.025, 0.050] | 0.003 [−0.002, 0.008] |
| Intercept | **1.598** [1.535, 1.658] | **2.972** [2.954, 2.993] | **4.066** [4.055, 4.077] | **0.243** [0.239, 0.247] |
| Observations | 6,500 | 6,500 | 6,500 | 6,500 |
| Model type (error distribution) | GLMM (binomial) | GLMM (negative binomial) | LMM (Gaussian) | LMM (Gaussian) |

[1]Although there is some correlation between predictor variables, particularly community size and density (S7 Fig), multicollinearity was reasonably low (max VIF = 3.0).

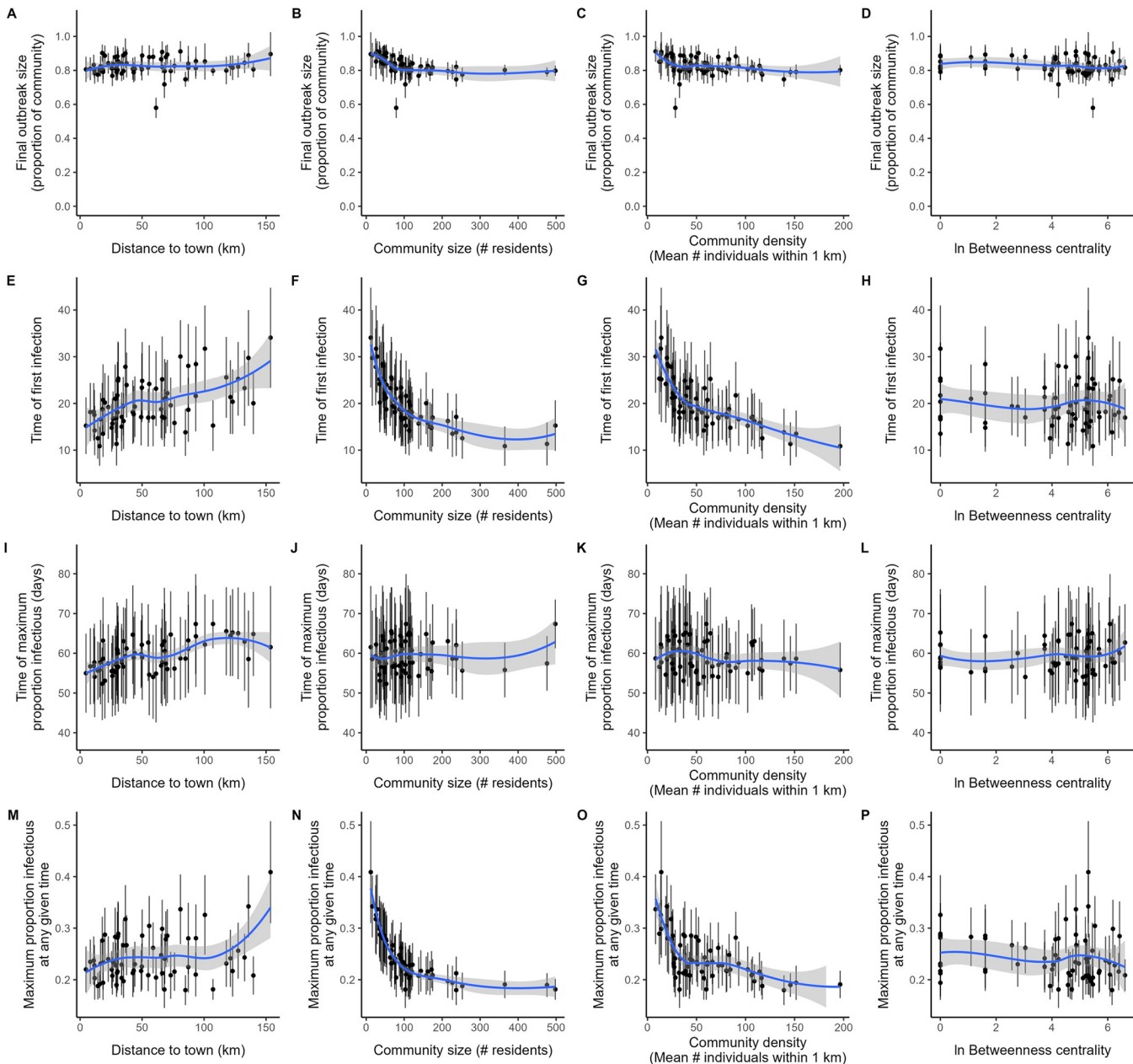

**Fig 4. Community-level predictors of infection risk. First row (A-D):** cumulative proportion exposed/infected; **Second row (E-H):** time step at which first infection was identified in community; **Third row (I-L):** time step at which the proportion of individuals actively infectious in a community reached a maximum during simulation; **Fourth row (M-P):** maximum proportion of individuals that were actively infectious at any time during simulation. Each outcome (row) is plotted as a function of community distance to nearest market town (first column), community size (second column), community density (third column), and ln+1-transformed community betweenness centrality (measured using an algorithm for weighted, directed graphs, with weights equal to the visitation probabilities from the travel sociomatrix used in the model) (fourth column). Points and intervals represent means ± SD. Smooths are unadjusted LOESS (locally estimated scatterplot smoothing) fits with span = 0.75. The data underlying this Figure can be found in http://doi.org/10.17605/OSF.IO/7YB2M (files: https://osf.io/jb5x7, https://osf.io/vtp3w).

## Potential interventions

Perturbations of model parameters revealed variability in the effectiveness of different intervention strategies in a Tsimane-like socioecology: restricting mobility, altering disease transmissibility, and restricting within-community gatherings.

**Table 3. Models of individual infection risk.** The GLMM included simulation, community, and individual as random intercepts, and model effects are presented as mean [95% CI]. Multilevel Bayesian hazard models included only a community random intercept and were run separately across simulations. Coefficients for M-splines describing the baseline hazard (with 7 df) are not shown. Posteriors were combined without weighting and effects are presented as median [95% CI]. Effects are bolded if CIs do not overlap zero.

| | Binomial GLMM (1) | Multilevel M-splines hazard model (2) |
|---|---|---|
| Intercept | **1.726** **[1.583, 1.869]** | **0.582** **[0.439, 0.714]** |
| Age | **0.098** **[0.085, 0.110]** | **0.046** **[0.010, 0.084]** |
| Sex (baseline = female) | **−0.230** **[−0.255, −0.205]** | **−0.090** **[−0.168, −0.018]** |
| Neighborhood density (# individuals within 1 km of focal) | **0.389** **[0.369, 0.410]** | **0.197** **[0.117, 0.285]** |
| Distance to center of community (km) | **−0.191** **[−0.208, −0.174]** | **−0.104** **[−0.181, −0.031]** |
| Adult household size | **0.731** **[0.715, 0.746]** | **0.316** **[0.261, 0.369]** |
| Community distance to town (km) | 0.020 [−0.102, 0.142] | −0.042 [−0.171, 0.097] |
| Ln-Community size (# of individuals) | **−0.367** **[−0.552, −0.182]** | **−0.219** **[−0.414, −0.033]** |
| Ln-Average community density | **−0.211** **[−0.392, −0.031]** | −0.071 [−0.256, 0.129] |
| Observations | 726,900 | 726,900 |
| Number of events | 587,806 | 587,806 |

**Restricting mobility.** Restricting travel to town, travel between villages, or all travel simultaneously by 50% had essentially no impact on disease outcomes at the population level (Fig 6A–6C). More severe travel restrictions (up to 90% reduction) either to town or between villages alone also had minimal impact on the cumulative proportion of infected individuals in the population by the end of simulation (Fig 6A and 6B). Extreme (90%) reductions in both town and between-community travel applied simultaneously were required to substantially slow transmission (approximately twice as long for epidemic to reach conclusion) and to reduce the overall proportion infected during the epidemic (approximately 15% reduction), a scenario that was also associated with much more variability across model runs (Fig 6C). Importantly, severely reducing town travel alone had little effect on final outbreak size but modified trajectories such that the timing of peak transmission was delayed; average time to reach 90% of total infections was 80 (95% percentile interval = [75, 85]) versus 108 [97, 118] days for the baseline versus 90% reduced travel, respectively (Fig 6A).

**Altered disease transmissibility.** In contrast, changes to transmissibility noticeably altered epidemiological outcomes. Doubling transmissibility led to 96% [95.8, 96.8] mean cumulative infection in the population, whereas halving transmissibility reduced the average cumulative incidence to 35% [31.3, 39.3] by the end of simulation (Fig 6D). Such reductions in transmissibility could be accomplished by the use of face coverings, vaccines, or novel disease variants. However, the fifth model scenario tested, in which transmissibility was altered solely at the disease source (town), demonstrated only a modest difference in disease trajectories and almost no difference in final outbreak size (Fig 6E). This suggests that efforts to encourage the use of facial coverings or increased caution when traveling to market towns and interacting with outsiders, absent of other interventions, is unlikely to be an effective mechanism of epidemic control in situations where the potential for rapid within-community spread is high.

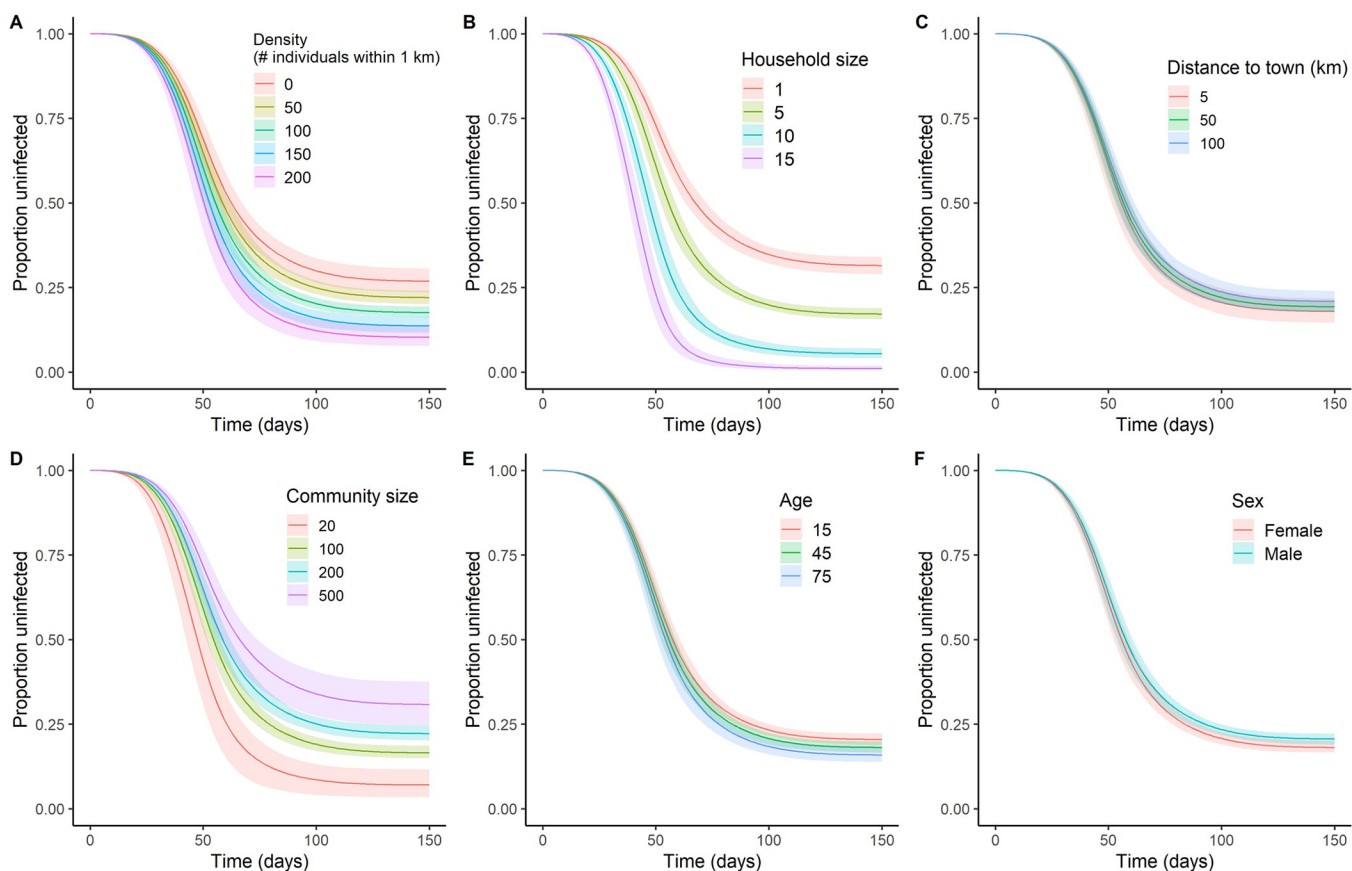

**Fig 5. Individual-level predictors of infection risk.** Cumulative proportion exposed/infected as a function of (**A**) neighborhood density (the number of individuals living within 1 km of focal), (**B**) household size, (**C**) distance to nearest market town (km), (**D**) community size (# individuals), (**E**) age (years), and (**F**) sex. The data underlying this Figure can be found in http://doi.org/10.17605/OSF.IO/7YB2M (files: https://osf.io/nmpfb, https://osf.io/2m4sp, https://osf.io/agyx7, https://osf.io/75u8w, https://osf.io/j25ku).

**Within-community gatherings.** Repeated local gatherings can substantially increase both the speed of outbreaks and final outbreak size (Fig 6F). Adding gatherings that mimic community meetings, church, or sporting events (occurring every 7 days), even relatively small events like parties or other social occasions (attended by 25% of the community) at this interval rapidly pushed the population to over 90% infected by the end of simulations. Higher attendance (50%) had an even stronger effect (Fig 6F). Although we did not explore scenarios combining the effects of gatherings and other interventions, it is likely that gatherings (which, in reality, are more frequent and occur for a variety of purposes, e.g., birthday parties) could offset any gains achieved by severe travel restrictions.

## Comparing simulations to real-world outcomes

Preliminary data on COVID-19 prevalence among Tsimane suggests that our baseline model accurately predicts empirical outcomes in this population, with an overall empirical adjusted [28] positivity rate of 81.1% (crude positivity rate = 75.7%) across communities following the first wave of infections [29]. In comparison, simulations predicted a mean cumulative incidence of approximately 80%. Likewise, the timing of the peak of observed positive cases occurred between approximately 40 and 60 days after initial infection [29], close to model predictions of 50 to 60 days. The model also accurately predicted a similar, but slightly female

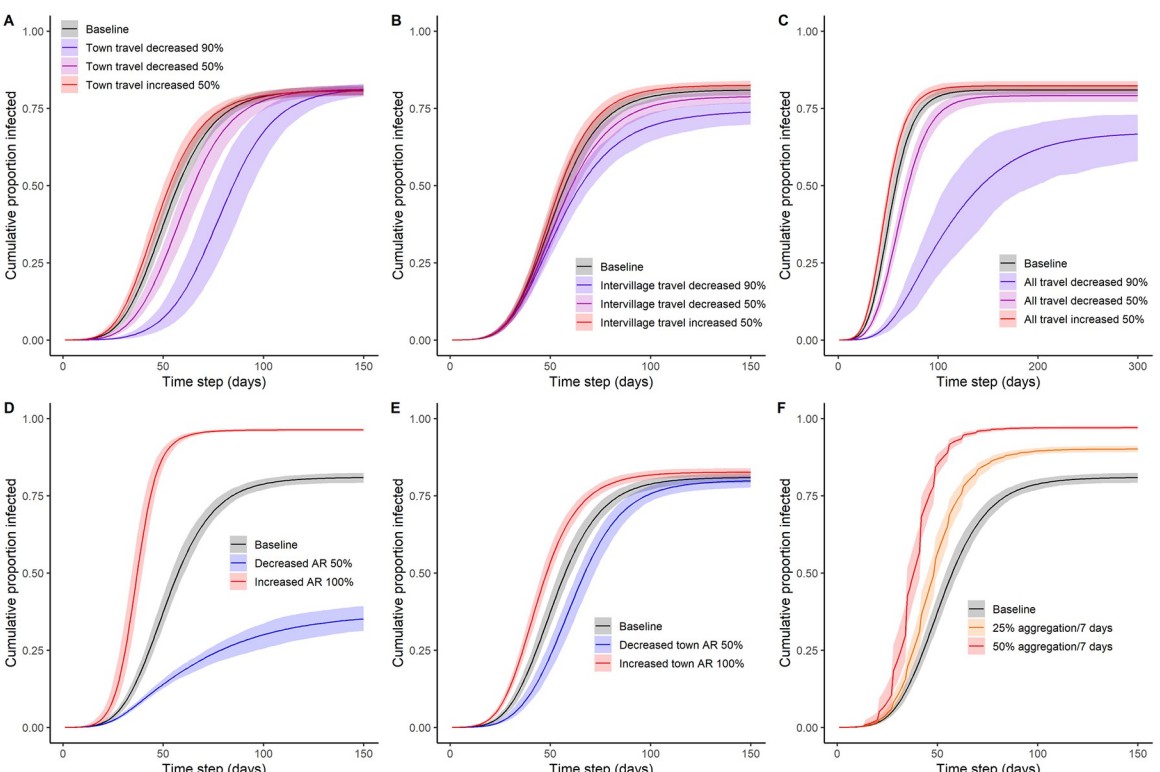

**Fig 6.** Epidemic trajectories of alternate model scenarios, with modified parameters for (**A**) travel to town, (**B**) intervillage travel, (**C**) both travel to town and intervillage travel, (**D**) attack rate (probability of infection spreading from infected to susceptible individual over 1 day of contact), (**E**) probability of contracting disease while visiting market town, (**F**) local aggregation events (with denoted percentage of community coming together at a regular time interval). All scenarios are depicted relative to baseline (black), with red-blue colors depicting degree of increase–decrease in target parameter. The data underlying this Figure can be found in http://doi.org/10. 17605/OSF.IO/7YB2M (files: https://osf.io/9qgpr, https://osf.io/57bm8, https://osf.io/gp8sn, https://osf.io/tewp4, https://osf.io/p8aqr, https://osf.io/4xdhc, https://osf.io/vp562, https://osf.io/42kxf, https://osf.io/anfhz, https://osf.io/hav6t, https://osf.io/xjubn, https://osf.io/h5fxe, https://osf.io/f798u, https://osf.io/a5cvm, https://osf.io/ykt52, https://osf.io/cer5h).

biased, cumulative incidence of infections between the sexes (observed adjusted seropositivity: male = 80.5% female = 81.5%; model: mean cumulative incidence male = 79.7%, female = 82.1%).

As a finer test of our predictions, we also compared model predictions to empirical cumulative incidence outcomes at the community level. The slope of the relationship between average model cumulative incidence and empirical crude seropositivity was close to 1 ($\beta \pm$ (SE) = 1.18 (1.87); Fig 7), demonstrating reasonable predictive power of our simulations. Likewise, the mean absolute deviation in community cumulative incidence (|simulated community mean–observed|) was 0.082, with an average deviation of −0.001, indicating that model errors were centered around zero. Interestingly, the largest outlier in this comparison (lowest point shown on Fig 7) represents a community for which social contacts data were available and was thus included in our statistical model used to estimate mean degree; examining the random effects structure of the empirical model reveals that this community had a strong, negative value of the random intercept term for cumulative contacts, indicating that individuals in that community interact much less frequently than those in an average community. This suggests that more fine-grained measures of the type reported here could help to further refine our model. Finally, as predicted by our model, communities located farther from market town did not have lower observed cumulative incidence compared to those in closer proximity to town (S5 Fig).

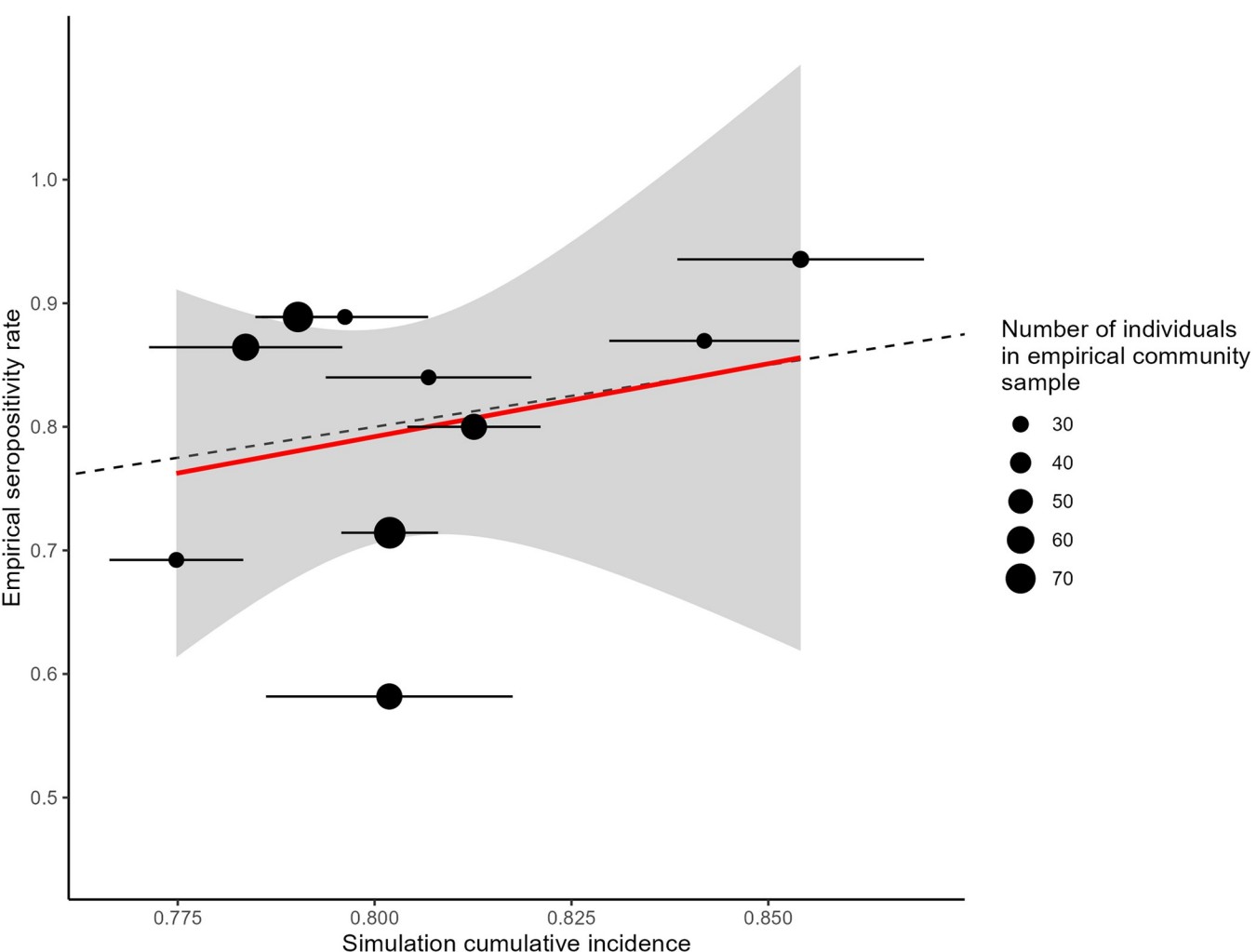

**Fig 7. Comparison of model predictions and empirical estimates of COVID-19 prevalence in Tsimane communities.** Communities were only included if they had a reasonable number of samples (>20). Points and intervals represent average model outcomes and bootstrapped 95% CIs, respectively. The size of points corresponds with the number of people sampled for SARS-CoV-2 antibodies in a given Tsimane community. Fitted line (solid, red) represents a weighted linear regression with weights equal to empirical sample sizes in each community. Dotted line represents a 1:1 relationship between model predictions and empirical outcomes. The data underlying this Figure can be found in http://doi.org/10.17605/OSF.IO/7YB2M (files: https://osf.io/3m7he, https://osf.io/jb5x7, https://osf.io/vtp3w).

## Discussion

Our empirically parameterized network model predicts a high potential rate of epidemic spread (approximately 80% cumulative incidence) for an infectious disease like COVID-19 entering a remote Indigenous community from an outside source, with a high degree of heterogeneity in the probability of becoming infected at the community and individual levels (Fig 2). At the community level, subpopulation size is the strongest predictor of total outbreak size and both the timing of first infection and the maximum instantaneous proportion of infections. Community density and proximity to town also contribute substantially to how quickly disease first reached a particular village (Table 2). At the individual level, infection is more likely among older adults, women, and individuals living in denser neighborhoods, larger households, and located more centrally in the community (Table 3). Tests of multiple intervention scenarios suggest that mobility restrictions (e.g., collective isolation) have little impact

unless behavioral change is rigidly enforced at extreme levels for both travel to town and between communities, whereas even minor reductions in transmissibility achieved through pharmaceutical and other interventions can effectively reduce disease spread and total outbreak size (Fig 6).

These findings suggest that rural Indigenous populations like the Tsimane exhibit both structural vulnerabilities and potential resiliencies against epidemics [6]. We next consider our results as they relate to each of our guiding questions.

## What features of Tsimane socioecology affect susceptibility to epidemics?

It is well known that historical epidemics in many Indigenous populations have had devastating effects due to widespread immunological naivety [2,30]. Less well understood, however, are the effects of geography, metapopulation structure, social organization, and other local characteristics. A study of the 1918 influenza epidemic, for example, found that factors beyond isolation and previous exposure were necessary to explain higher mortality in aboriginal communities, including higher concurrent infectious disease rates, greater crowding, lower genetic diversity, and poor access to primary care [4].

The model developed here provides a theoretical basis for considering how socioecological factors govern susceptibility to epidemics, particularly as Indigenous communities become more integrated into market economies. We find that the interconnected metapopulation structure of communities—as represented by a series of increasingly remote Tsimane villages spread along rivers and roads—reduces the extent to which geographical distance alone facilitates isolation and protection. For example, simulated outbreaks spread in a chain-like fashion; once the disease is introduced, high rates of travel to nearby communities encourages proliferation along a local gradient (S3 Fig). Analogous to species persistence in classic ecological models, metapopulation structure with migration can affect disease persistence and the susceptibility of differently sized local populations in complex ways [31,32]. In our model-based simulations, these dynamics may account for the inability of any single community to avoid severe outbreaks across simulations.

Additionally, analysis of the empirical data underlying our model helps explain high rates of simulated infections. Daily contact with other community members estimated from observational social network data is frequent and heterogeneous (low homophily) by age and sex, unlike in industrialized populations where formal institutions tend to structure interactions [33]. The combination of intergenerational mixing and large numbers of within-household contacts may promote transmission of infectious disease by large droplets or small droplet nuclei, as has also been shown in South African townships [34]. Although interactions among Tsimane are biased towards genetic and affinal kin and those living in close proximity, this bias is not strong enough to offset high connectedness across communities. Likewise, travel to market towns and between villages is frequent (mean empirical probabilities of intervillage travel and travel to town are 9% and 4% of days, respectively). Although young men were the most likely to travel to town, travel rates are high enough across age-sex classes that meaningful mobility restrictions would have to target large swaths of the population to be effective. In sum, once a SARS-CoV-2-like virus enters an Indigenous population with similar socioecological characteristics as the Tsimane, viral spread is likely to occur quickly due to cultural and behavioral factors such as high connectivity and mobility, communal living, and shared meals/households [35].

Human populations worldwide are also resilient via local strategies for coping with epidemics. One well-documented example is the response of central Africans to Ebola outbreaks, which included the abandonment of areas known to be disease epicenters, local prohibitions

against travel in and out of villages, mandated quarantine protocols, and a shift away from normal cultural practices such as large funerals [36]. Reviewing the ethnographic literature, McGrath [37] reported a range of common responses to epidemics that are likely ancient, most notably fleeing, migration, and quarantine/isolation measures. We concur with others that interventions building off of such existing local mechanisms are the most likely to be successful [36].

The Tsimane exhibit notable resiliency in several ways. For example, many families maintain separate houses located near their horticultural fields, farther away from village centers. They often inhabit those houses during the wet season rice harvest, but also during periods of conflict or turmoil. Movement into these residences during disease outbreaks could potentially be leveraged to reduce population densities and flatten epidemic trajectories. Tsimane subsistence production also affords self-sufficient individuals the ability to survive independent of store-bought market resources and associated contact risks [6]. Although this is changing with increasing market integration [38], and although we are skeptical of the protection afforded by isolation per se, the Tsimane are nonetheless capable of distancing themselves from broader social contacts in a way that other populations are not. It is therefore possible that highly effective voluntary collective isolation could be achieved when cultural models of disease react to a threat that is perceived to be extremely dire, as observed during Ebola outbreaks in Africa [36].

### Individual and community risk factors

Our results also have implications for how medical resources might be distributed in certain remote populations in advance of an impending epidemic. Overall, our simulations demonstrated only moderate heterogeneity in both the timing and magnitude of peak infection within communities (Fig 2B). We found that outbreak size was proportionally largest in the smallest communities and that first and peak infections occur earliest in communities that are near market towns (Figs 2 and 4 and Table 2). Perhaps surprisingly, our empirical model of daily contacts indicated that community size has minimal effect on contact rates, with a slight but highly uncertain trend towards a higher number of unique daily contacts in smaller communities. Overall network density is therefore higher in smaller communities if mean degree is similar across communities of different size. The importance of local density as a key driver of individual infection risk (Table 3) in metapopulations also concords with a large body of existing work [39,40], but our high overall infection outcomes challenge the conclusion of Li and colleagues [39] that epidemics are necessarily limited in relatively low-density populations.

Contrary to our intuitions, proximity to town and community centrality (betweenness) had little apparent effect on the magnitude of community infections (Table 2). Our results suggest that maximum public health impact may be achieved by focusing limited medical and messaging resources not on large, dense communities close to town where an infectious disease is most likely to arrive first, but rather preferentially towards small outlying communities where outbreaks are likely to be most severe. Alternatively, predictions of community trajectories suggest that mobile medical resources during an incipient outbreak could be distributed according to a combination of village location and size and then redistributed dynamically as villages farther from town reach peak infection.

Driven by heterogeneity in empirically estimated microlevel processes that govern individual network structure in the Tsimane population (e.g., individual attributes, travel proclivities, and spatial layout), our model makes testable predictions about factors likely to elevate individual risk of infection in an epidemic. In order of decreasing importance, these include larger household size, higher neighborhood density, greater proximity to the center of a community, female sex, and older age (Table 3). People who live close to the village center may also be

more likely to participate in aggregations or events typically held in these locations, and thus the observed effect could be exacerbated under more realistic conditions. Although men were at a slightly lower risk than women, likely due to men having a marginally lower mean network degree, the magnitude of this effect was small (Fig 5) and the greater mobility of men means that sex plays an important role in initial disease introduction. These results concord with findings in similar systems demonstrating the effects of intense within-household mixing and contact [41] and suggest that spatial density can play an important role in epidemics and may thus help guide public health efforts.

The coupling of detailed modeling with empirical sampling of disease outcomes in the same population lends further insight into the drivers of transmission in this system. Overall, comparison suggests that baseline predictive performance of the model was quite good (Fig 7) using baseline parameters representative of SARS-CoV-2 (Table 1) and long-term data on social interaction and mobility in Tsimane communities. Tracking network evolution and epidemic spread over time reveals a straightforward path by which large communities near market towns experience early introductions of infection, followed by rapid spread within and between close neighboring communities. Sequential spread driven by intercommunity travel ensues, rendering even the most remote communities vulnerable to outbreaks; an unfortunate outcome mirrored in empirical observations of Tsimane communities located on a gradient from approximately 10 to 70 km from the nearest market town (S5 Fig). In addition, we can understand a small female bias in infection rates in both model and empirical outcomes in light of social network data showing that women have a slightly larger number of daily contacts, but with a lack of strong sex homophily in those interactions that precludes the development of a large overarching disparity. Finally, community-level comparisons, particularly where observed outcomes do not match expectation, reveal areas for theoretical model improvement. For example, and as noted earlier, the outlier in Fig 7 may be explained by community heterogeneity in contact rate and social network structure due to unexplained factors, suggesting an important target of future research that could help elucidate protective mechanisms that do not rely on top-down intervention. The strength of the individual-based network modeling approach demonstrated here, as compared to a deterministic compartmental model of disease transmission, is that our model naturally accommodates metapopulation structure, mobility dynamics, and social network structure along with stochasticity to yield predictions about heterogeneity in infection outcomes; understanding such heterogeneity is critical to addressing our motivating questions regarding whether certain community or individual subgroups are at greater risk for infection during an epidemic.

## Intervention strategies: Is voluntary collective isolation effective?

Remoteness is an epidemiological double-edged sword. On the one hand, remoteness can prevent the introduction of disease into communities altogether, such as the approximately 19% of Alaskan populations that avoided exposure to the A(H1N1) virus in 1918 to 1919 [42]. On the other hand, a history of contact can reduce immunological naivety and thus decrease mortality rates when epidemics reach Indigenous populations [4]. Faced with a globally novel virus like SARS-CoV-2 where all populations were initially naïve, we advocated for voluntary collective isolation as part of a broader mitigation strategy for the Tsimane, given their potential for self-sufficiency, relative lack of available personal protective equipment or vaccines, and limited access to medical facilities [6], and many Indigenous communities worldwide implemented similar isolation protocols [11,13].

To test the theoretical efficacy of voluntary collective isolation, we ran model simulations to test for effects of restricted travel to town, between communities, or both. We found that these

mechanisms achieved minimal effect on the magnitude of disease outbreaks unless implemented at a severe level (Fig 6). Critically, our results suggest that intervention success requires reductions of both travel outside of Tsimane territories and visitation to other Tsimane communities. The latter is much more difficult to control using a top-down approach and likely to be more difficult to achieve socially.

Models of highly disparate systems have demonstrated that reduced travel in metapopulations can delay the timing of peak infections but are unlikely to affect final epidemic size unless implemented with high efficacy early in outbreaks [43,44]. Indeed, models of crude historical records of individuals from rural Canada during the 1918–1919 influenza epidemic were used to demonstrate the inefficacy of quarantine as a protective strategy for isolated communities under all but the most extreme conditions [45], in agreement with the results we present here. Partial isolation therefore does not appear to be effective at reducing outbreak severity, although it may prove useful for slowing the regional spread of an epidemic to relieve pressure on medical resources. Indeed, relatively insulated populations like the Amish in the United States appear to have experienced COVID-19 mortality rates similar to the broader US population [46]. Given that complete isolation is unlikely achievable for Tsimane or many other rural subsistence populations, our results challenge the efficacy of the collective isolation approach and suggest that modern medical resources, perhaps via establishment of rural health posts that serve villages directly, must be directed to remote communities. Isolation should thus be viewed as a strategy to delay transmission until sufficient medical resources are available rather than a path to complete protection [6], as underscored by recent outcomes in China following the end of a governmental "zero-COVID" policy [47].

More broadly, available evidence suggests that among Indigenous populations, voluntary collective isolation is undermined by the following: (1) conditions that promote the majority of disease spread occurring between and within communities after introduction, and not contingent on repeated arrivals from regional urban centers (Fig 3); (2) a high degree of autonomy and lack of top-down control, where villagers "vote with their feet," making enforcement difficult; and (3) changing socioeconomic conditions that make interaction with outsiders critical to local livelihoods. Extreme isolation has been effective in preventing the arrival of epidemics in some instances in the past, such as Arctic populations during the 1918 influenza [4,42], but contemporary conditions are not comparable in the extent of isolation. The simulations presented here based on Tsimane data—combined with the context in which most Indigenous populations live today—provide further evidence that controlled contact with relatively isolated populations, mediated by cultural experts and health professionals, may be critical to avoiding the decimation of remaining isolated Indigenous communities [48].

To date, we have observed a variety of self-isolation procedures by Tsimane to limit the spread of COVID-19 into communities, including blockading roads to limit travel [6]. These actions mirror reports of numerous attempts by Indigenous communities worldwide to implement travel restrictions into their territories to limit the spread of COVID-19 [11], but in most, if not all, cases, data suggest that such measures have ultimately proved unsuccessful in either urban or remote Indigenous communities [13]. This includes the Tsimane case, where we have observed limited sustained adherence to community isolation protocols. This outcome is likely a product of continuing incursion into Indigenous territories by outsiders (miners, lumber harvesters, truck drivers, etc.) combined with the need for Indigenous peoples to engage in trade and commerce to sustain their livelihoods. A further complication is that many Indigenous communities still lack territorial sovereignty, thereby hindering efforts to control or enforce travel within traditional territories [49]. The results of our model support these suppositions and demonstrate formally the difficulties associated with suppressing a pandemic via collective isolation in conditions lacking top-down oversight.

Alternatively, and as expected based on other studies, reductions in transmissibility are predicted to be more effective in curtailing disease spread (Fig 6) [43]. Unfortunately, a focus on reduced transmission while visiting market towns was found to have little effect, and thus changes (social distancing, face coverings) would have to be implemented within communities (Fig 6E). We do note, however, that behavioral interventions to reduce transmissibility may be difficult to implement among Tsimane and similar populations; the extent to which people share food, household items, and living quarters can be intense, and there is a lack of access to personal protective equipment. Public health messaging might instead be better focused on encouraging individuals to avoid unnecessary between-household contacts and traditional gatherings that could lead to superspreader events. This may be particularly effective, given that reducing large gatherings is much more feasible compared to reducing intrahousehold contacts, food sharing with neighbors, or other common forms of interaction.

## The future of epidemics in subsistence populations

Nearly all Indigenous populations today, including those relying heavily on subsistence farming or hunting and gathering, maintain relationships with out-groups and vary along a spectrum from autarky to high embeddedness in market interactions. Exposure to epidemics is typically initiated through direct contact with outsiders in regional or local hubs or via incursions into local territories [50]. With increasing market integration, susceptibility to exposure will be influenced by several processes. First, improved infrastructure and travel technology (e.g., new roads, outboard motors, motorbikes) can increase urban–rural disease spread by enabling more frequent trips to market towns for trading goods, shopping, or socializing [18]. Travel technology can also increase the frequency of intercommunity travel, reducing the extent to which distance limits exposure in the most remote communities. Second, the arrival of loggers, miners, and merchants reduces the protection afforded by isolation. Although in some cases such incursions are illegal or imposed by outsiders [11], improved access to economic opportunities and goods are welcomed by many Indigenous communities. Potential mitigation strategies must therefore be realistic about individual incentives to comply with public health messaging. Third, high rates of population growth and environmental degradation may lead to increased reliance on market goods and longer travel in search of resources. Finally, religious or government-sponsored aggregations (e.g., church, school) can increase clustered contacts and potentially amplify disease spread (Fig 6). Increasing market integration, especially when combined with lack of access to healthcare, is therefore expected to elevate the susceptibility of most Indigenous populations to exposure from epidemics. It will also elevate reverse processes of disease transmission from populations that interact frequently with wildlife to urban centers. The creation of roads, schools, stores, and economic development should be matched with access to modern medical facilities and/or rural health posts, even in communities that may appear to be relatively isolated, in order to avoid infectious disease catastrophes.

The mitigation of harmful epidemics among relatively isolated populations in the future will depend to some degree on the successful deployment of vaccines. Clear challenges in this arena include not only the logistics of distribution amid limited infrastructure (e.g., no or poor roads, lack of proper storage facilities such as −80°C freezers for mRNA vaccines, lack of local clinics) but also mistrust and misinformation that cause hesitancy and limited uptake. Indeed, after 3 years into the COVID-19 pandemic, few Tsimane have received vaccines. Our conversations with local communities suggest an overwhelming reticence due to fear and poor public health messaging. Because vaccine hesitancy in remote-living Indigenous groups like the Tsimane may have a different etiology than comparatively urban, industrialized populations,

there is a tremendous need to identify successful models of uptake in these contexts for future application.

## Future directions

A limitation of this study is the omission of children less than 10 years old. We excluded young kids for 2 main reasons. First, given the large size of our study population, computational demands (both in terms of simulation time and available RAM) limited our ability to conduct simulations on a greater number of individuals. Second, we have less detailed information on model parameters relating to visitation, mobility, and social networks of children in this population. Although the extent to which children play an important role in SARS-CoV-2 transmission remains poorly understood, children have the potential to further increase intravillage connectedness via playing, visiting, or through organized activities such as schooling. As dependents, children often accompany their parents when traveling to town or between villages and thereby may exacerbate transmission between locations. The inclusion of children would also increase household sizes and community densities, potentially exacerbating the rate at which disease spreads. Our model could thus be refined by including young individuals as well as aggregations in those age classes for villages that have established schools.

## Conclusions

Like many Indigenous populations, the Tsimane exhibit demographic, social network, and mobility patterns that differ from urban industrialized populations. Combined with the on-the-ground reality of limited public health resources and cultural barriers, these factors challenge the applicability of standard epidemiological models to make detailed predictions about disease spread in small-scale rural metapopulations. Using a stochastic network model parameterized with rich empirical data, we generated population-specific predictions about the spread of a SARS-CoV-2-like virus in an Indigenous population of Amazonian forager-horticulturalists under different conditions and found that their relative isolation is unlikely to offer substantial protection from novel epidemics due to a combination of high mobility, intervillage travel, and dense contact networks within communities. Voluntary collective isolation is only likely to be successful with an unusually high degree of top-down control or community buy-in and with equal restrictions on travel to market towns and between villages. Public health messaging focused on reducing transmissibility within communities and protecting older adults and other vulnerable individuals, either by vaccination or by social distancing, should be prioritized as in urban industrialized contexts. Authorities should also plan to distribute medical resources to even the most remote-living communities. Finally, this study illustrates how data collected by social scientists—censuses, longitudinal data on geospatial positioning and residence, time allocation, behavior, demography, and mobility—can be marshalled for epidemiological purposes to improve our ability to respond to future epidemics in a greater diversity of societies.

## Materials and methods

### Study population

Our model employs extensive empirical data collected among Tsimane Indigenous Amerindians (population approximately 17,000) inhabiting the vicinity of the Maniqui and Quiquibey river systems in the Beni Department of Bolivia. There are over 90 distinct Tsimane communities in the region that range in size from approximately 50 to 500 people [27]. Subsistence is derived from a combination of shifting horticulture (cultigen staples are sweet manioc,

plantain, rice, and corn) and foraging (i.e., fishing, hunting, and gathering forest foods), with cooperative production and extensive sharing within and between families [51]. Relatedness within communities tends to be high and visitation with kin between communities occurs frequently [52].

Until recently, Tsimane have had limited access to modern healthcare and the market economy due to the relatively remote location of villages. Important changes over recent decades have begun to alter the extent to which Tsimane interact with non-Tsimane Bolivians (*napo*) and markets. Road building, timber extraction, internal migration, missionization, and the introduction of technologies that increase mobility (e.g., motorboats) are key factors that have led to increased market integration and cultural change. Their impact varies in proximity to the local towns of Yucumo (population: approximately 5,000), Rurrenabaque (population: approximately 20,000), and San Borja (population: approximately 42,000), as residents of villages located closer generally travel to town more often.

Empirical data derive from 2 decades of research on demography, behavior, health, and life history by the Tsimane Health and Life History Project (THLHP) [27]. The THLHP operates a mobile medical team that travels between villages in conjunction with targeted research campaigns including biomedical researchers and anthropologists. Census records are collected during community visits regarding village residents and visitors. Long-term demographic data provide information on the age, sex, and kin relations of individuals. We used this information to construct a starting population for our model including 7,269 individuals (excluding all individuals aged <10) with known age and sex in 65 villages covered by THLHP research. Individuals aged <10 were excluded due to computational constraints for simulations (memory and time requirements) and evidence suggesting that children are less susceptible to and play a lesser role in transmission of SARS-CoV-2 [53,54].

Since 2007, the THLHP collected household interviews with GPS to characterize the composition and spatial layout of communities. GPS data were available for approximately 63% of individuals in the current study. We imputed missing spatial data by sampling from the kernel density of communities with greater GPS coverage (S1 Text).

## Inclusivity in global research

Additional information regarding the ethical, cultural, and scientific considerations specific to inclusivity in global research is included in S1 Checklist.

## Epidemiological model

**Dynamic, stochastic individual network modeling using TERG models.** We used the R package *EpiModel* (version 2.3.1, 53) in the *Statnet* suite to model the dynamics of infectious disease transmission in a metapopulation of Tsimane community networks with empirically parameterized demographics, migration, and contact rates (Fig 1). The *EpiModel* package uses the temporal exponential-family random graph model (TERGM) framework [55] to simulate discrete-time dynamic contact networks in a defined population. This is achieved by using a statistical model of interaction based on empirical knowledge of the microlevel processes that govern social contact (tie) formation and dissolution. In brief, a population is constructed based on the node-level characteristics of individuals in a target sample (see description below). Egocentric network data on microlevel social processes are used to generate target statistics that describe system characteristics (e.g., average degree, node/tie attributes, homophily, other network statistics). These target statistics are used in combination with the sample to fit a TERG model describing the dynamics of tie formation/dissolution using Markov Chain Monte Carlo maximum likelihood estimation (MCMC-MLE). Because TERG models are

generative, they can be used to simulate networks in which outcomes vary stochastically around the target statistics [56]. Once a TERG model is defined, disease transmission parameters (Table 1) and custom modules (see below) are set that govern additional behavior or demographic processes. Finally, dynamic simulations are run with tracking at the individual level for analysis of disease transmission dynamics. For further details of the approach, see [57,58].

The *EpiModel* package contains an application programming interface that allows for customizable extension of base models. We employ several custom extensions to accommodate the metapopulation structure of Tsimane communities, travel between communities and to nearby market towns, and SEIRD (susceptible-exposed-infectious-recovered-death) dynamics.

SEIRD dynamics are classically implemented such that individuals can be in one of 5 compartments at each time step: susceptible, exposed, infectious, recovered, or dead. Exposed individuals are infected but not yet contagious, and recovered individuals are immune to further infection. Based on a daily probability of (mean infection duration)$^{-1}$ (here, the tunable mean infection duration parameter is set to 10 days; Table 1), infectious individuals either survive (transition to a recovered state) or die according to age-specific case fatality rates [59].

At the outset of our model, all individuals are susceptible, and travel to a nearby market town is the only source of potential infection. The infection risk profile when travelling to town is specified as a truncated normal distribution with a theoretical maximum of 0.05 (5% probability of being infected after 1 day spent in town) with start and end at days 0 and 60, respectively (S4 Fig). This town infection profile is set to reflect the fact that nearby towns likely experienced a wave of infection spanning approximately 2 months (reducing town infection to 0 after this period ensures that the full dynamics of the epidemic play out without continuous seeding). A normal distribution was chosen because it approximates the empirical distribution of active COVID-19 cases reported for the country of Bolivia during the initial outbreak in summer 2020 (https://covid19.who.int/region/amro/country/bo).

Travel between communities and to market towns is governed by a module that dynamically tracks the location of all individuals. At each time step, the model starts by determining whether an individual will (1) remain in her home community, (2) move to town, or (3) or visit another Tsimane community, by sampling from a vector of travel probabilities determined by the age, sex, and home community of each individual (see S1 Text for details of the empirical data and models used to generate travel probability vectors). Individuals that move out of their home communities to visit another Tsimane community are assigned to a random household in the visitation community (and the associated geographical distance matrix associated with being in that household) and have all existing network connections removed from the last time step (to ensure no between-community ties).

**Baseline scenario.** For all scenarios, we ran 100 simulations over 150 days. The input parameters for our baseline model are detailed in Table 1. Tie formation was restricted to occur within-communities and tie dissolution was set to 1 day, such that ties are resimulated based on target parameters at each step of the model (S1 Text). Model inputs are either derived from empirical data on Tsimane or published reference values relating to the SARS-CoV-2 virus. As such, this scenario serves as a baseline case to generate expectations about transmission dynamics from underlying low-level processes. Next, we explore how interventions or behavioral changes that modify model parameters would affect epidemiological outcomes in this population.

**Intervention strategies.** Several modifications to model parameters were made to explore how potential interventions or behavioral changes would affect epidemiological outcomes.

1. **Travel to town:** Alter the frequency with which individuals travel to market towns. This could represent introduced technology (e.g., motorboats that increase the frequency of

visits) or interventions encouraging villagers to avoid travel that would increase contact with non-Tsimane Bolivians.

2. **Intervillage travel:** Alter the frequency with which individuals travel to nonresident Tsimane communities. Generally, intervillage travel represents social visitation or travel seeking economic opportunities.

3. **All travel:** Alter the frequency with which individuals travel to both town and other Tsimane communities.

4. **Within-community attack rate:** Alter the probability of transmission between susceptible and infected contacts within Tsimane communities. This parameter could be altered by social distancing, facial mask usage, or SARS-CoV-2 variants that spread more easily.

5. **Town transmissibility:** Alter the probability of transmission during visits to market town.

6. **Local aggregations:** Dynamically modify within-community networks by creating local aggregations specified to occur at a given frequency (i.e., how often aggregations form on a recurring basis) and intensity (i.e., what proportion of the community aggregates, and what percentage of dyads form ties). Tsimane communities regularly engage in such aggregations, including during community meetings or celebrations, church or other religious services, school attendance, and soccer games.

## Empirical data

**Microlevel social processes.** Data representing Tsimane contact networks were derived from a large behavioral observation database collected in 8 communities between 2002 and 2007. Time allocation data were collected as periodic scan samples taken every 30 minutes in 2- to 3-hour time blocks between 7 AM and 7 PM within a focal housing cluster. For each person scan, an anthropologist recorded all individuals present in the social group (defined as participating either in the same conversation [actively or passively] or within 3 meters of proximity), the current activities they were engaged in, and all individuals in the same activity group (defined as participating in the same cooperative endeavor). If an individual was resident in a time block but not present at the time of sampling, other residents were interviewed about the missing person's whereabouts and activities outside of the household cluster (follow-up interviews indicated a high degree of reporting accuracy). Limiting the time allocation data to individuals who were residents of the focal clusters and age 10+ to match our model population yielded a total of 44,781 scan samples from 681 unique individuals ($n_{men} = 358$, $n_{women} = 323$, mean (range) age: 29 (10 to 84) years).

The TERGM framework was used to simulate longitudinal social networks based on target statistics derived from these observational data. To generate target statistics, we created 2 summary datasets. In the first, we generated rows that summarized across all observation points for each unique person-day. For each person-day, we calculated the total unique number of alters encountered in the same social group (degree), the number of those unique alters that fell within each of 5 age categories (10 to 25, 26 to 40, 41 to 55, 56 to 70, and 70+), the average age difference between ego and all alters, the proportion of alters that were male, average genetic relatedness between ego and all alters, average affinal relatedness between ego and alters, and the average ln-transformed distance (in meters) between households of ego and all alters. Each person-day was thus treated as an egocentric network sample from which we then calculated weighted averages to use as target statistics (with weights equal to the degree). The second summary dataset was used to estimate the daily mean degree for men and women by

age group. Mean degree is fundamental to the model because it determines the number of contacts at each time point that can lead to infectious disease transmission. Because time allocation data were collected at 30-minute intervals within time blocks that did not cover the span of an entire 24-hour day, we could not directly estimate the cumulative number of unique daily contacts directly. We therefore calculated the number of cumulative unique alters encountered by ego in a single day based on all observations of ego. For example, if person A had contact with alters B and C at time point 1, and B, C, and D at time point 2, the cumulative unique degree would be assigned values of 2 and 3, respectively. From these data, we then estimated a Bayesian multilevel power-law model with random slopes using the *brms* package in R of the form:

$$D_i \sim Poisson(\lambda_i)$$

$$\begin{aligned}
\log(\lambda_i) = &\ \alpha_{individual[k]} + \gamma_{day[j]} + \beta_0 male_i + \beta_1 community\ size_i + \beta_2 age[26-40]_i \\
&+ \beta_3 age[41-55]_i + \beta_4 age[56-70]_i + \beta_5 age[70+]_i + \rho_{day[j]} \ln(ntmbk)_i
\end{aligned}$$

$$\begin{bmatrix} \gamma_{day} \\ \rho_{day} \end{bmatrix} \sim MVNormal(\begin{bmatrix} \gamma \\ \beta \end{bmatrix}, \mathbf{S})$$

$$\mathbf{S} = \begin{pmatrix} \sigma_\gamma & 0 \\ 0 & \sigma_\beta \end{pmatrix} \mathbf{R} \begin{pmatrix} \sigma_\gamma & 0 \\ 0 & \sigma_\beta \end{pmatrix}$$

$$\mathbf{R} = \begin{pmatrix} 1 & \rho \\ \rho & 1 \end{pmatrix}$$

$$\beta_{0-5} \sim Normal(0, 0.3)$$

$$\alpha, \gamma \sim StudentT(3, 0, 1.5)$$

$$\sigma_\beta, \sigma_\gamma \sim StudentT(3, 0, 2)$$

$$\mathbf{R} \sim LKJcorr(1)$$

where $D_i$ is the unique cumulative degree for observation $i$ of individual $k$ on day $j$, *male* is the sex of an individual (1 = male, 0 = female), *community size* is the number of individuals living in the community where the focal is resident, and *ntmbk* is the sequential number of observation time blocks for individual $k$ on day $j$ (numbered from 1, 2, 3, ..., $j$). We then used the resulting model to estimate the predicted daily number of contacts at *ntmbk* = 24 (12 hours/ d = 24 30-minute observation points) by sex and age for an average individual. Community size was set to the average for predictions because the effect size of this variable was extremely small ($\beta$ [95% CI] = −0.07 [−0.11, −0.03]).

**Visitation and travel.** There are 2 types of empirically parameterized travel in the model. The first is travel to local Bolivian market towns, the potential initial source of infection. To estimate the frequency of travel to towns, we utilized interview data collected during routine medical visits between 2006 and 2018 ($n$ = 7,874 observations of individuals age 10+) in which participants were asked, "How many days did you spend in town last month?" and "Which town did you visit?" Respondents reported spending between 0 and 28 days per month in

town (mean = 1.5). We fit a generalized linear mixed model (GLMM) with a Poisson error distribution for the number of days per month spent in town as a function of age, age$^2$, sex, and sex-by-age and sex-by-age$^2$ interactions, with random intercepts for community of residence and individual. This model was used to generate daily predicted probabilities of travel to town for each of the 7,269 individuals in the simulated population based on age, sex, and home community.

The second type of travel included in our model is movement between the 65 communities ("intervillage"), a common feature of Tsimane life that is shared with many other small-scale societies worldwide. Intervillage travel was parameterized using data derived from comprehensive surveys of individual travel histories collected in 2010 to 2011 ([52]; see S1 Text; S6 Fig).

## Data analysis

At the community level, we employed multilevel generalized linear models (GLMMs) to assess the effect of independent variables (community size, density, betweenness centrality, distance to town) on the proportion of individuals infected, time of first infection, the time at which the maximum proportion of individuals were infectious, and the maximum proportion of individuals that were infectious within a community. Proportion infected was modeled with a binomial error distribution with number of successes and failures representing the number of individuals infected or not infected, respectively, by the end of simulation. Time of infection was modeled using a negative binomial error distribution (overdispersed count variable), and the remaining timing variables were fit with Gaussian models. All models incorporated a random intercept term for community (repeated across simulations).

We additionally assessed individual probability of infection by fitting a multilevel generalized linear model with a binomial error distribution to data collated across all simulations ($n$ = 726,900 person-simulations). Each observation represents a single individual within a simulation, with infection status by the end of the simulation as a (binary) outcome and individual- and community-level attributes as predictors. Additional individual attributes include age, sex, household size, neighborhood density (number of individuals living within 1 km of focal's house), and distance to center of community (location where central meetings and aggregations most commonly occur, usually near church, school, or community meeting spot). Crossed random intercepts were included for individual, community, and simulation.

Finally, we used Bayesian multilevel hazard models to characterize individual probabilities of infection over time. To do so, we modeled a flexible baseline hazard function using M-splines in addition to fixed (individual and community predictors) and random (community intercept) effects using the "stan_surv" function in the *rstanarm* package. Whereas the GLMMs described above assess the probability of an individual node ever being infected over the course of simulation as a function of predictor variables, the parametric hazard models assess time to infection event outcomes and are used to generate the survival curves in Fig 5. Models were fit and standardized survival curves were generated from each simulation separately at different levels of predictor variables, before being combined to calculate mean and 95% intervals across simulations.

## Ethics statement

This research was approved by institutional review boards (IRBs) at the University of New Mexico (#07–157) and the University of California, Santa Barbara (#3-21-0652). In addition, data collection was approved by and discussed with the Tsimane government (Gran Consejo Tsimane), local community leaders, and study participants. Informed consent was obtained in

either Tsimane or Spanish (choice of participant) from all participants who contributed empirical data. Individual-level empirical data on the Tsimane population used as model input in this study are highly identifiable (GPS locations of households, individual household sizes, ages, and sexes, etc.) and thus are not publicly available.

## Supporting information

**S1 Checklist. Inclusivity in global research questionnaire.**
(DOCX)

**S1 Text. Supplementary materials and methods text.**
(DOCX)

**S1 Table. Description of the generative exponential family random graph model fit to empirical data and used to simulate contact networks across the Tsimane population.**
(DOCX)

**S1 Fig. Histograms of total outbreak size by community across all 100 simulation runs.** Communities are ordered by increasing size. Outcomes here are for simulations under baseline conditions. The data underlying this Figure can be found in http://doi.org/10.17605/OSF.IO/7YB2M (files: https://osf.io/jb5x7, https://osf.io/vtp3w).
(TIFF)

**S2 Fig. Animation of individual transmission dynamics in a single community taken from a single baseline model run.** Individuals are arranged by the relative geographic location of households, slightly jittered for visualization. When individuals travel to town or other communities, their nodes move to the boxes on the right, with labels indicating current location. Colors indicate infection status (blue = susceptible, black = exposed, red = infectious, green = recovered, deaths = none occurred). Community infection is seeded at time 17–18 when an individual visits town and is exposed. An interactive version of this figure is available online at https://thomaskraft.github.io/epidemic_simulations/fig_s2.html
(MP4)

**S3 Fig. Dynamic visualization of epidemic spread across Tsimane communities from 4 example model simulations.** The fluctuating size of bubbles reflects movement in and out of resident communities (e.g., numbers represent individuals in a community at a given time point). https://thomaskraft.github.io/epidemic_simulations/landscape_spread_tsimane.gif.
(GIF)

**S4 Fig. Town infection profile exhibiting dynamics of a single wave in the source population with a maximum effective attack rate of 0.05.** This profile assumes that it was possible to contract the disease in town over 2 months (60 days) from a single symmetric "wave," peaking at a maximum 5% probability of contracting the disease (marked with red line). The data underlying this Figure can be found in http://doi.org/10.17605/OSF.IO/7YB2M (files: https://osf.io/m892d).
(TIFF)

**S5 Fig. Empirical SARS-CoV-2 seropositivity rate by community as a function of distance to the nearest market town.** Each point represents a single community in which >20 seroassays were conducted. The data underlying this Figure can be found in http://doi.org/10.17605/OSF.IO/7YB2M (files: https://osf.io/3m7he, https://osf.io/jb5x7, https://osf.io/vtp3w).
(TIFF)

**S6 Fig. Map visualizing community interconnectedness.** The thickness and transparency of edges is proportional to visitation probabilities in the baseline model. Yellow circles represent villages (size proportional to population size), and blue circles represent major market towns. Villages are displayed roughly according to location in geographic space.
(PNG)

**S7 Fig. Correlation matrix of community-level predictor variables (*n* = 65 communities).** The upper triangle shows Pearson correlation coefficients (95% CIs), and the lower triangle shows bivariate scatterplots. Variance inflation factors for variables in community models are generally low with a maximum of 3.0 (community size–community density), indicating that multicollinearity is unlikely to be a problem for inference given large effect sizes and sample size. The data underlying this Figure can be found in http://doi.org/10.17605/OSF.IO/7YB2M (files: https://osf.io/vtp3w).
(TIFF)

## Acknowledgments

We are grateful to all Tsimane participants, staff, and the Tsimane Gran Consejo for making this work possible. We also thank the organizers of the "Understanding and Exploring Network Epidemiology in the Time of Coronavirus" workshop series hosted by the University of Maryland's COMBINE program (NSF #1632976) and the University of Vermont's Complex Systems Center, which originally inspired this research. Use was made of computational facilities purchased with funds from the National Science Foundation (CNS-1725797) and administered by the Center for Scientific Computing (CSC). The CSC is supported by the California NanoSystems Institute and the Materials Research Science and Engineering Center (MRSEC; NSF DMR 2308708) at UC Santa Barbara. Martina Morris and Chad Klumb provided crucial technical support, and Susan Cassels provided helpful comments and advice.

## Author Contributions

**Conceptualization:** Thomas S. Kraft, Edmond Seabright, Sarah Alami, Michael D. Gurven.

**Data curation:** Thomas S. Kraft, Edmond Seabright, Paul Hooper, Bret Beheim.

**Formal analysis:** Thomas S. Kraft, Edmond Seabright, Samuel M. Jenness, Bret Beheim.

**Funding acquisition:** Hillard Kaplan, Michael D. Gurven.

**Investigation:** Thomas S. Kraft, Edmond Seabright, Sarah Alami, Paul Hooper, Bret Beheim, Helen Davis, Daniel K. Cummings, Daniel Eid Rodriguez, Maguin Gutierrez Cayuba, Emily Miner, Xavier de Lamballerie, Lucia Inchauste, Stéphane Priet, Benjamin C. Trumble, Jonathan Stieglitz, Hillard Kaplan, Michael D. Gurven.

**Methodology:** Thomas S. Kraft, Edmond Seabright, Samuel M. Jenness.

**Project administration:** Thomas S. Kraft, Hillard Kaplan, Michael D. Gurven.

**Software:** Thomas S. Kraft.

**Supervision:** Hillard Kaplan, Michael D. Gurven.

**Visualization:** Thomas S. Kraft.

**Writing – original draft:** Thomas S. Kraft.

**Writing – review & editing:** Thomas S. Kraft, Edmond Seabright, Sarah Alami, Samuel M. Jenness, Paul Hooper, Bret Beheim, Helen Davis, Daniel K. Cummings, Daniel Eid

Rodriguez, Maguin Gutierrez Cayuba, Emily Miner, Xavier de Lamballerie, Lucia Inchauste, Stéphane Priet, Benjamin C. Trumble, Jonathan Stieglitz, Hillard Kaplan, Michael D. Gurven.

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
