## [Editor Report · Decision Letter 0]

3 Apr 2023

Dear Dr. Kraft, 

Thank you for submitting your manuscript entitled "Metapopulation dynamics of infectious disease (SARS-CoV-2) transmission in a small-scale human society" for consideration as a Research Article by PLOS Biology.

Your manuscript has now been evaluated by the PLOS Biology editorial staff, as well as by an academic editor with relevant expertise, and I am writing to let you know that we would like to send your submission out for external peer review.

Once your full submission is complete, your paper will undergo a series of checks in preparation for peer review. After your manuscript has passed the checks it will be sent out for review. To provide the metadata for your submission, please Login to Editorial Manager (https://www.editorialmanager.com/pbiology) within two working days, i.e. by Apr 05 2023 11:59PM.

Kind regards,

Paula

---

Senior Editor

PLOS Biology

---

## [Decision Letter · Decision Letter 1]

8 Jun 2023

Dear Dr Kraft,

Thank you for your patience while your manuscript "Metapopulation dynamics of infectious disease (SARS-CoV-2) transmission in a small-scale human society" was peer-reviewed at PLOS Biology. It has now been evaluated by the PLOS Biology editors, an Academic Editor with relevant expertise, and by two independent reviewers. I'm handling this paper temporarily on behalf of my colleague Dr Paula Jauregui, while she is out of the office.

Based on the reviews, we are likely to accept this manuscript for publication, provided you satisfactorily address the remaining points raised by the reviewers. Please also make sure to address the following data and other policy-related requests.

IMPORTANT - please attend to the following:

a) Please could change your Title to "Metapopulation dynamics of SARS-CoV-2 transmission in a small-scale human Amazonian society"

b) Please read our "Inclusivity in Global Research" policy (https://journals.plos.org/plosbiology/s/best-practices-in-research-reporting#loc-inclusivity-in-global-research) and download, complete and upload the questionnaire.

c) Please address the requests from reviewers #1 and #2.

d) Please attend to my Data Policy requests below; specifically, we need you to supply the numerical values underlying Figs 2AB, 3, 4ABCDEFGHIJKLMNOP, 5ABCDEF, 6ABCDEF, 7, S1, S4, S5, S6, S7, either as a supplementary data file or as a permanent DOI’d deposition. I note that you already have an associated GitHub deposition (https://github.com/ThomasKraft/tsimane_epidemic), but this URL does not appear to work; please rectify this. We're aware that the raw data cannot be shared because of privacy concerns, but please could you clarify whether the data/code are sufficient to reproduce your Figures? Also, because Github depositions can be readily changed or deleted, please make a permanent DOI’d copy (e.g. in Zenodo) and provide this URL (see below).

e) Please cite the location of the data clearly in all relevant main and supplementary Figure legends, e.g. “The data underlying this Figure can be found in S1 Data” or “The data underlying this Figure can be found in https://doi.org/10.5281/zenodo.XXXXX”

We expect to receive your revised manuscript within two weeks. 

*Published Peer Review History*

*Press*

Sincerely,

Roli Roberts

Roland G Roberts PhD

Senior Editor

PLOS Biology

rroberts@plos.org

on behalf of

Editor,

pjaureguionieva@plos.org,

PLOS Biology

DATA POLICY:

Regardless of the method selected, please ensure that you provide the individual numerical values that underlie the summary data displayed in the following figure panels as they are essential for readers to assess your analysis and to reproduce it: Figs 2AB, 3, 4ABCDEFGHIJKLMNOP, 5ABCDEF, 6ABCDEF, 7, S1, S4, S5, S6, S7. NOTE: the numerical data provided should include all replicates AND the way in which the plotted mean and errors were derived (it should not present only the mean/average values).

DATA NOT SHOWN?

REVIEWERS' COMMENTS:

Reviewer #1:

This paper discusses the spread of COVID-19 and its severity in indigenous populations. The author considers an individual-based network stochastic model instead of a traditional compartmental model as a novel approach as these indigenous communities are relatively small in population size and isolated from urban centers. Overall, the paper is well written, and the results section provides community information and individual-level COVID-19 transmission, mobility, and an excellent comparison of the effects of human behavioral changes on infectious numbers.

Comments:

1. There is a lack of information about the two models (Binomial GLMM, Multilevel M-splines hazard model in Table 1) used in the Results: outbreak section. I would suggest briefly describing why you choose the two models in the result section.

2. The animation Fig. S2 runs only until t=5-6, but in the paper, you talk about the disease dynamics at t=17-18. Please fix this (in the Total outbreak size - Community level Section).

3. It is confusing that the dependent variables in Table 2 are referred to as models throughout the paper (e.g.: Table 2: Model 3, Fig. 4i-l). Replace "model" with "column" or "variable".

4. Please use one format only (uppercase or lowercase, but not both) when naming the figures and when citing them in the context. Right now, the figures are in uppercase, and they are referred to in paragraphs with lowercase letters.

5. Please cite this information "mean probabilities of intervillage travel and travel to town = 9% and 4% of days, respectively" (in Discussion Section paragraph 6).

6. I suggest changing the method of transportation: "boat motors" to "motorboats".

7. I believe the TERGM framework in the Empirical data section is same as the STERGM framework discussed in the results section, if it is not, please mention it clearly.

8. More explanation is required for the four sub-figures in animation Fig. S3, I do not see much difference among them. In addition to that, it would be helpful lowering the play speed of the animation and provide a pause/play button(s) with the animation.

9. Please fix the link for the data and code (https://github.com/ThomasKraft/tsimane_epidemic), which is not working at the moment.

Reviewer #2:

In this paper, the authors simulated the SEIRD transmission model on a stochastic social network of Indigenous communities in Amazon. The network has 7269 nodes and is parameterized with high-resolution empirical data on population structure, mobility, and contact networks. 

This is a well-written paper. The results are interesting and consistent with the simuation results. The methods employed are explained in a clear and concise manner, making them easily comprehensible for readers. The flow of the paper is good, I believe the readers can effortlessly navigate through its contents. Personally, I found reading the paper to be a pleasant experience.

Upon careful examination, I couldn't identify any significant flaws in the paper. However, I did come across a few minor issues after conducting thorough checks.

 I recommend the publication of this paper in PLOS Biology if the suggestions are considered. 

1. Page 30. The authors should explain why 'a truncated normal distribution with a theoretical maximum of 0.05 ($5\\%$ probability of being infected after one day spent in town)' is being used. Is the infection risk follow normal distribution? Why is it a normal distribution?

2. Page 34. In the second formula, age[10-26] is missing. 

3. Page 48. Figure 4. A typo 'loess' in the last sentence.

4. Page 56. The Figure S2 in the listed website doesn't match the description in paper.

---

## [Editor Report · Decision Letter 2]

17 Jul 2023

Dear Dr Kraft,

Thank you for the submission of your revised Research Article "Metapopulation dynamics of SARS-CoV-2 transmission in a small-scale Amazonian society" for publication in PLOS Biology. On behalf of my colleagues and the Academic Editor, Jason Ladner, I am pleased to say that we can in principle accept your manuscript for publication, provided you address any remaining formatting and reporting issues. These will be detailed in an email you should receive within 2-3 business days from our colleagues in the journal operations team; no action is required from you until then. Please note that we will not be able to formally accept your manuscript and schedule it for publication until you have completed any requested changes.

PRESS

Sincerely, 

Paula 

---

Senior Editor

PLOS Biology
